# Evaluation of the Effectiveness of a Phytogenic Supplement (Alkaloids and Flavonoids) in the Control of *Eimeria* spp. in Experimentally Challenged Broiler Chickens

**DOI:** 10.3390/ani15060847

**Published:** 2025-03-15

**Authors:** Anne-Sophie Hascoët, Paulina Torres-Celpa, Roberto Riquelme-Neira, Héctor Hidalgo-Olate

**Affiliations:** 1Technical Department, MPA Veterinary Medicines and Additives (Grupo PH-Albio), 08210 Barcelona, Spain; 2Laboratorio de Patología Aviar, Facultad de Ciencias Veterinarias y Pecuarias, Universidad de Chile, Santiago 8820808, Chilehhidalgo@uchile.cl (H.H.-O.); 3Facultad de Medicina Veterinaria y Agronomía, Universidad de Las Américas, Santiago 8242125, Chile; riquelmeneiraroberto@gmail.com

**Keywords:** poultry, *Eimeria* spp., anticoccidial drugs, drug resistance, phytogenic supplements

## Abstract

Coccidiosis is a costly disease in poultry farms worldwide caused by microscopic intestinal parasites. Resistance to conventional drugs is a growing problem, making alternative solutions essential. In this study, chickens were experimentally infected with *Eimeria* spp. to test the effectiveness of a natural plant-based supplement compared to commonly used drugs. Chickens exposed to the parasitism and given the plant-based supplement or drugs had fewer coccidia oocysts in their feces and less intestinal damage compared to untreated chickens. The plant-based supplement showed comparable effects to anticoccidial drugs in controlling coccidiosis, without affecting growth, feed intake, or survival. This suggests that plant-based supplements could be a natural and effective alternative to drugs for managing coccidiosis. Developing such options may help reduce drug resistance and improve the health of chickens, benefiting both farmers and consumers. Further research is needed to confirm its long-term effectiveness and use in real-scale farming conditions.

## 1. Introduction

A significant amount of money is spent globally on the prevention, control, and treatment of poultry coccidiosis. Concretely, Blake et al. (2020) estimated the global cost of coccidiosis in chickens to have been approximately EUR 11.91 billion in 2016, considering performance losses and control costs (prevention and treatment expenses) [1]. This ubiquitous disease is mainly caused by a protozoan parasite belonging to the phylum Apicomplexa, family Eimeriidae, genus *Eimeria* [2]. This parasite infects the intestinal epithelium over a period ranging from 4 to 6 days [3]. Depending on the severity of the infection, *Eimeria* spp. can cause a wide range of lesions, from mild to severe [2]. These lesions can lead to nutrient malabsorption, weight loss, ruffled feathers, bloody diarrhea, and, in extreme cases, death [4]. Eradicating coccidia is particularly challenging due to the formation of resilient oocysts, which can persist in the environment and resist commonly applied disinfectants [5].

There have been numerous anticoccidial drugs and vaccines used to prevent and control this parasite [6]. Most anticoccidial drugs, such as monensin and amprolium, inhibit the growth, metabolism, and reproduction in different stages of the coccidia life cycle. However, due to their widespread use, some of these drugs have been associated with the development of resistance to their active principle [5,7,8,9,10,11,12,13]. An antimicrobial agent is a substance that can either kill or inhibit the growth of microorganisms. Both antibacterials and anticoccidials are classified as antimicrobials, with antibacterials targeting bacteria and anticoccidials specifically designed to combat coccidia [5]. The development of multiple resistance in field isolates can occur for various reasons, including changes in cell membrane permeability which prevent the drug from being absorbed or cause it to be rapidly expelled from the cell. Other factors include the use of alternative biochemical pathways, alterations in the target sites of coccidia, and genetic recombination [5]. Resistance to coccidiostats and other antimicrobials in poultry production is a growing and concerning issue. Several studies have documented noteworthy cases of resistance in *Eimeria* spp. and other associated pathogens. For instance, Ferdji et al. (2022) reported total resistance to monensin and robenidine, as well as partial resistance to salinomycin and the combination of narasin and nicarbazin in *Eimeria* spp. oocysts isolated from a poultry farm [14]. Similarly, Flores et al. (2022) identified severe resistance to multiple coccidiostats, including clopidol, diclazuril, maduramicin, monensin, salinomycin, and toltrazuril, in chickens infected with nine different field samples [8]. Resistance to toltrazuril has been particularly highlighted in experimental studies. Ojimelukwe et al. (2018) documented resistance to this compound in experimentally infected chickens [13], while Lan et al. (2017) reported severe resistance to toltrazuril, sulfonamides/trimethoprim, and amprolium [12]. Furthermore, the use of conventional coccidiostats has been linked to increased resistance in *Campylobacter* spp. to other antimicrobials. Hansson et al. (2021) reported a higher incidence of *Campylobacter jejuni* resistant to quinolones in systems applying coccidiostats [15]. Similarly, Avrain et al. (2003) found that the application of coccidiostats increases *Campylobacter* spp. resistance to antimicrobials such as ampicillin and tetracycline, as well as the incidence of *Campylobacter coli* [11].

These findings underscore the need for integrated and sustainable management strategies to mitigate resistance, not only in *Eimeria* spp. but also in bacterial pathogens that may compromise animal health and food safety. Acid-based products, probiotics, prebiotics, symbiotics, yeast-based products, plant-derived substances, combinations of these, and other similar products, have been developed and introduced to the market as supplements for animal feed [16,17,18,19,20,21]. The use of botanicals and their secondary metabolites for their antimicrobial and antiparasitic properties dates back centuries, but their potential role as an anticoccidial agent is relatively new [22,23]. The effectiveness of these products against coccidia has been tested with inconsistent, conflicting, or unconvincing results. Mohiti-Asli et al. (2015) found that oregano oil (500 ppm) reduced oocyst shedding similarly to diclazuril, indicating an interesting anticoccidial effect [24]. In contrast, Giannenas et al. (2004) reported that oregano (5–7.5 g/kg) reduced lesion scores but was less effective than lasalocid, with higher doses (10 g/kg) potentially causing toxic effects [25]. These results suggest that oregano oil can help control coccidiosis, but its efficacy varies depending on dosage and its comparison with drugs. Despite the availability of botanicals with anticoccidial properties, their effectiveness varies across studies and needs further validation under controlled conditions. Some of these products seem to target the bacterial microbiota rather than directly affecting coccidia [5]. Others achieve this by preventing and controlling the development and progression of coccidiosis in poultry, primarily through direct interference with the life cycle of *Eimeria* spp. or indirectly by enhancing host immunity, restoring gut morphology, maintaining microbial homeostasis, and balancing oxidative and antioxidant status. Some examples of plants tested in poultry production include *Artemisia annua*, *Aloe secundiflora*, *Bidens pilosa*, *Curcuma longa*, *Dichroa febrifuga*, *Emblica officinalis*, *Allium sativum*, *Camellia sinensis*, *Origanum vulgare*, Radix Sophorae Flavescentis, *Saccharum officinarum*, and *Yucca* [26,27,28]. Interestingly, it can be observed that more plants with bioactive compounds, such as polyphenols, tannins, and alkaloids, have been tested [26,27,28].

The present study was designed to evaluate the anticoccidial efficacy of a phytogenic supplement of alkaloids and flavonoids in broilers artificially infected with a mix of six strains of *Eimeria* spp. and to compare its effects on broiler performance and intestinal health with commonly used anticoccidial additives.

## 2. Materials and Methods

### 2.1. Animals

A total of 256 one-day-old Ross 308 male broilers were randomly distributed into 4 groups with 8 cages each (8 chicks per cage). The study was conducted using a Completely Randomized Design. The chicks were reared under standard conditions in experimental stainless-steel battery brooders in an isolated room. The batteries were equipped with an integrated heating system to regulate the temperature following the recommendations for this broiler genetic line [29], gradually decreasing from day 1 to day 23 of age. Humidity was not actively controlled. Lighting followed a human work schedule, with darkness at night and additional artificial light during the day. Each cage was equipped with a feeder, a water cup, and an individual floor waste tray to collect excreta. The experimental period lasted 23 days. A commercial standard diet, formulated without anticoccidials and based on a corn and soybean meal, was used to meet the nutritional requirements for the broilers according to the specifications of the US National Research Council [30]. Each diet was prepared at the production facilities of the Facultad de Ciencias Veterinarias y Pecuarias—Universidad de Chile, and the birds had ad libitum access to water and feed. The four experimental groups were as follows: (1) negative control (NC): standard diet without anticoccidials and no *Eimeria* spp. challenge; (2) positive control (PC): standard diet with *Eimeria* spp. challenge but no anticoccidials; (3) phytogenic combination supplement (PhCS): standard diet without anticoccidials, supplemented with bioactive compounds at 1 kg/ton of feed during the whole experiment, and with *Eimeria* spp. challenge; (4) anticoccidial (NN): standard diet formulated with narasin and nicarbazin (Maxiban™ by Elanco, containing 80 g/kg of narasin and 80 g/kg of nicarbazin, dosed at 0.6 kg/ton of feed, during all experimental periods) and with *Eimeria* spp. challenge (Table 1). The feed supplement used in the PhCS group was Eimex^®^, which contains 10% isoquinoline alkaloids and 10% flavonoids as active ingredients. It was provided by MPA Veterinary Medicines and Additives—Grupo PH-Albio (Barcelona, Spain).

### 2.2. Coccidia Oocysts Challenge

On day 14, the positive control and PhCS and NN groups were challenged with an oral overdose of a mixture containing 20 times the standard dose (4.0 × 10^4^ *Eimeria* spp. oocysts/bird), following the standardized methodology of the laboratory at the Universidad de Chile adapted from Grenier et al. (2016) [31]. The mixture included 1.2 × 10^4^ *E. acervulina*; 0.4 × 10^4^ *E. brunetti*, *E. maxima*, and *E. tenella*; 0.8 × 10^4^ *E. mivati* and *E. necatrix* live oocysts (Coccivac^®^-D2, MSD Animal Health, Santiago, Chile).

### 2.3. Oocysts Count

Daily, from day 1 to 9 days post-inoculation (PI), the number of oocysts per gram of feces was examined in samples collected from each cage (8 cages per experimental group). A modified McMaster counting chamber technique was used, following a standardized protocol from the Laboratorio de Patología Aviar of the Facultad de Ciencias Veterinarias y Pecuarias of the Universidad de Chile and inspired by the modification described by Wetzel (1951) [32,33,34,35]. Briefly, each fecal sample (100 g) was homogenized with 50 mL of water to achieve a uniform suspension of the oocysts. A total of 0.6 g of homogenized feces was suspended in 60 mL of saturated sodium chloride solution (density 1.2 g/cc) and stirred for two minutes with a vortex mixer. This created a dilution factor of 1:100. After dilution, the suspension was used to fill six McMaster chambers per sample, if necessary, due to high oocyst content. The number of oocysts per gram of feces was calculated using the formula (mean oocyst count under 6 columns × 100)/6). The result was expressed as the number of oocysts per gram of feces (OPG). An average was calculated from the eight counts obtained for each experimental group.

### 2.4. Intestinal Lesion Assessment

On day 9 post-inoculation, all birds were euthanized by cervical dislocation for the macroscopic observation of intestinal lesions (duodenum, jejunum-ileum, and cecum) according to the method previously described by Johnson and Reid (1970) [36]. A numerical ranking of gross lesions was performed using a discrete 5-step scale (0 = no lesions; 1 = mild lesions; 2 = moderate lesions; 3 = severe lesions; 4 = extremely severe lesions or death). According to Conway and McKenzie (1999), scoring was carried out by the same person throughout the study [37]. For the analysis, the scores for each intestinal section were aggregated to obtain a single value per chicken (maximum score of 12 points) as follows: total intestinal lesion score = duodenum score + jejunum/ileum score + cecum score.

### 2.5. Histopathological Analysis

For histopathological analysis, one chick per replicate (cage) was randomly selected, resulting in a total of 8 chicks per experimental group. The analysis focused on the villus/crypt ratio (villus height divided by crypt depth) in the duodenum and jejunum, as well as the height of the cecal mucosa. The samples were previously fixed in 10% neutral buffered formalin (Sigma-Aldrich Química, Santiago Chile) and processed for histological examination using routine hematoxylin and eosin staining and were analyzed under an optical microscope (Olympus CX31, Olympus, Center Valley, PA, USA, with 10× eyepieces and 20× objective lens). Samples were evaluated for coccidial distribution and severity according to the methodology described below, and these scores were summed to produce a total score reflecting coccidial tissue damage. The total score for intestinal histopathology was composed by the “coccidial” score (distribution and severity) and the “inflammation” score. The total histopathology score was calculated as follows: Total histopathology score = (Coccidiosis distribution score + Coccidiosis severity score) + Inflammation score. The coccidial distribution score (based on distribution in 4 fields at 10× magnification) was determined as follows: 0 = no coccidia; 1 = coccidia in 1 field; 2 = coccidia in 2 fields; 3 = coccidia in 3 fields; 4 = coccidia in all 4 fields. The coccidiosis severity score (percentage of affected villi in 4 fields at 10× magnification) was determined as follows: 0 = no coccidia; 1 = <25%; 2 = 25–50%; 3 = 50–75%; 4 = >75%, as indicated by Idris et al. (1997) [38]. The inflammation score was classified as follows: 0 = no microscopic lesions; 1 = mild infiltrate; 2 = extensive infiltrate and edema; 3 = extensive infiltrate and edema extending to the muscularis mucosa; 4 = degeneration of glands, necrosis, and hemorrhage, as described by Sultan et al. (2019) [39]. Finally, at least 10 random fields at 40× magnification were examined before a sample was considered negative (score 0 for each parameter).

### 2.6. Performance Evaluation

For performance data, all broilers in each group were weighed on days 1 and 23. Feed intake per cage was recorded from day 0 to 23. The average daily weight gain (ADWG) and feed conversion ratio (FCR) were calculated as the ratio between feed intake and weight gain per cage. In addition, mortality was monitored daily throughout the entire experiment. Clinical signs (such as diarrhea, watery or pasty droppings, loss of appetite, pale comb and wattles, lethargy, hunched posture, and stunted growth) were also checked daily, and necropsies were performed to observe macroscopic lesions whenever necessary.

### 2.7. Statistical Analysis

All data were presented as the mean ± 5% error using the SAS system (version 9.3, SAS Inst. Inc., Cary, NC, USA). Data were analyzed qualitatively, through the description of the variables, and quantitatively, previously performing the normality test (Shapiro–Wilk for days 5, 6, 8, and 9 and D’Agostino and Pearson for day 7) (*p* > 0.05) and using a one-way analysis of variance (ANOVA) test followed by Tukey’s post hoc multiple comparison test to determine statistical differences between experimental groups. A *p*-value < 0.05 was considered statistically significant. Oocyst shedding analysis was based on cage means, and oocyst counts were logarithmically transformed (log10(OPG)) to meet normality assumptions before statistical analysis for days 6, 7, and 9.

## 3. Results

### 3.1. Oocyst Count

The effects of the phytogenic supplement and anticoccidial additives on *Eimeria* spp. OPG are shown in Figure 1. The first fecal oocyst excretion was detected on day 5. Additionally, as expected, no oocysts were observed in the negative control group during the analyzed period. The maximum OPG was observed on day 6 for all coccidial-challenged groups, with the PhCS and NN groups having approximately 20,000 fewer OPG counts than the positive control. Both the phytogenic supplement and anticoccidial additives show a numerical reduction in OPG counts between days 6 and 9 post-inoculation compared to the positive control, with significantly lower counts in the PhCS group on days 6 and 9 (*p* < 0.05). In fact, on day 9, PhCS and NN showed a 61.3% and 71.6% reduction in oocyst count, respectively, compared to the positive control.

### 3.2. Intestinal Lesions

In the intestinal lesions assessment at 9 days post-infection, no lesions were observed in the negative control group, as expected. Furthermore, no significant differences were observed in the average intestinal lesion scores between the challenged groups (PC: 4.36/12; PhCS: 4.13/12; NN: 4.21/12) (*p* = 0.217) (Figure 2).

### 3.3. Histopathology

As expected, the NC group did not present microscopic lesions at day 9 post-inoculation (Figure 3, Table 2). The lesions observed in the PC group were significantly greater than those detected in the NC and the NN groups (*p* < 0.05). The PhCS group obtained an intermediate average total histopathology score (5.125/12) (Figure 3, Table 2), with no significant differences compared to either the PC (9.75/12) or NN (2.875/12) (*p* > 0.05).

In general, a sparse-to-moderate number of coccidian organisms was observed in the studied samples from the positive control and PhCS and NN groups, with these findings being relatively scarce when considering the total number of samples. The developmental forms identified—schizonts, microgametes, and macrogametes—were present in all samples from these groups. These forms were located in the epithelium; however, in the positive control group, they were occasionally observed in the lamina propria. The inflammatory response ranged from minimal in the negative control and NN groups to mild in the positive control and PhCS groups. Inflammation was mainly characterized by heterophils, an increased number of lymphocytes in the lamina propria, or lymphocytes undergoing epithelial transmigration. No samples showed necrosis, hemorrhage, edema, or infiltration by leukocytes other than those mentioned above. Mild epithelial and glandular hyperplasia was frequently observed, characterized by an increased number of layers, an irregular epithelial contour, and numerous mitotic figures in the crypts (glands). These changes were usually associated with the presence of coccidia in the positive control and PhCS groups. In the cecum, mild sporadic heterophilic inflammation was observed without the presence of coccidia (NN group). In these samples, filamentous bacteria were found on the surface and particularly within the crypt lumens.

### 3.4. Performance

At the end of the trial day (day 23), body weight did not differ significantly between groups (*p* = 0.151) (Figure 4), with an average weight of 800.5 ± 97 g. No treatment effect was found on feed intake (*p* = 0.966), with an average feed intake of 993.8 ± 28.9 g; or on the feed conversion ratio (*p* = 0.123), with an average FCR of 1.31 ± 0.36. There was no mortality in any group, and inoculated broilers exhibited only mild or absent clinical signs of coccidiosis.

## 4. Discussion

The search for alternatives to anticoccidial drugs is a crucial area of research in poultry production. As highlighted by Nguyen et al. (2021), the emergence of drug-resistant *Eimeria* spp. strains underscores the necessity for alternative solutions [40]. In the present study, a mixed-species *Eimeria* spp. challenge in poultry was conducted to compare the efficacy of a phytogenic supplement (PhCS) which combines isoquinoline alkaloids and polyphenols against a standard anticoccidial diet supplementation.

When broilers are exposed to coccidial pathogens, their intestinal integrity is often compromised, leading to significant economic losses due to reduced performance and poor health outcomes. Fecal oocyst excretion is widely recognized as a key parameter for evaluating coccidial infection [2,21]. Numerous reports have documented the efficacy of plant-derived anticoccidials and natural products in controlling coccidiosis [10,19,20,41,42]. Consistent with these findings, the results of the present study demonstrate that PhCS effectively reduced fecal oocyst excretion in birds infected by coccidia from the sixth day post-challenge. In fact, flavonoids have been shown to target the asexual phase of the protozoa, reducing both intestinal invasion and fecal oocyst shedding [42,43]. In this study, no significant differences were observed between groups regarding macroscopic intestinal lesions at 9 days post-inoculation. Since the lesions caused by *E. tenella* and *E. maxima* appear between 3- and 6-days post-inoculation, it might have been beneficial to conduct an intermediate evaluation by slaughtering a representative sample of chickens during this period [44,45]. Additionally, Landi-Librandi et al. (2012) further explained that the beneficial effects of polyphenols in parasitized animals are related to their ability to reduce oxidative stress as part of the host’s defense mechanism [46]. Similarly, Nweze and Obiwulu (2009) demonstrated that flavonoids derived from *Ageratum conyzoides* at a rate of 1 kg/ton of feed reduced both oxidative stress and intestinal lesions caused by *E. tenella* in poultry [16]. Reinforcing our findings of reduced microscopic intestinal inflammation, flavonoids—known to contain aromatic hydroxyl groups—exhibit antioxidant properties [47] that alleviate inflammation [48] and enhance the host’s defense against infections and oxidative stress. These anti-inflammatory, antioxidant, and antiparasitic properties [42,49] contribute to minimizing oxidative stress [46], limiting intestinal damage caused by *Eimeria* spp. and promoting beneficial bacterial growth and immune system activation [16].

Alkaloids have shown significant potential as natural antibiotics, with broad-spectrum antibacterial activity, minimal side effects, and a low risk of drug resistance [50]. For instance, sanguinarine induces apoptosis in *E. tenella* sporozoites via reactive oxygen species, reduced mitochondrial membrane potential, and increased calcium ion concentrations [51]. Additionally, isoquinoline alkaloids provide immunomodulatory effects, such as reducing proinflammatory cytokine interleukin-8 in hens with Spotty Liver Disease [51]. That effect could be relevant for managing various pathologies. Berberine, another alkaloid, also exhibits anticoccidial effects, reducing fecal oocyst excretion in most *Eimeria* species. Nguyen et al. (2021) have evaluated the anticoccidial effects of berberine-supplemented diets in broilers challenged separately with different *Eimeria* species (*E. acervulina*, *E. maxima*, *E. tenella*, *E. mitis*, and *E. praecox*). This bioactive compound was found to reduce fecal oocyst excretion in most *Eimeria* species, although it was less effective against *E. maxima* [40]. While assessing the impact on each species separately is valuable, the present study’s approach—which evaluates multiple *Eimeria* species simultaneously—may provide a more representative view of the birds’ real microbiota, where diverse microbial species coexist. In fact, as indicated by Flores et al. (2022), who conducted an epidemiological study on *Eimeria* in chicken farms in Korea, 77.5% of the positive samples contained between three and five *Eimeria* species, with the most prevalent being *E. acervulina* (98.6%), *E. maxima* (84.8%), and *E. tenella* (82.8%) [8]. Berberine has also been shown to enhance growth performance in yellow-feathered broilers, comparable to antibiotics like oxytetracycline [52]. This improvement was linked to changes in the cecal microbiota, including increased phylum *Bacteroidetes* and the genus *Bacteroides*, alongside reduced *Firmicutes* and genera like *Lachnospiraceae* and *Intestinimonas* [52]. Similarly, isoquinoline alkaloids influence the *Firmicutes/Bacteroidetes* (F/B) ratio in a dose-dependent manner [53].

As highlighted by Othman et al. (2019), both polyphenols and alkaloids have significant potential as secondary metabolites to act as resistance-modifying agents, offering a viable strategy to mitigate the spread of bacterial resistance [54]. In this study, the combination of alkaloids and polyphenols exhibited notable anticoccidial activity, suggesting that their spectrum of action could be broader, comparable to that of ionophores [5]. However, unlike ionophores, these natural compounds appear to have a lower propensity for inducing resistance, further emphasizing their importance as a therapeutic alternative in combating resistant pathogens [50].

About performance metrics, weight gain serves as a sensitive and informative measure of anticoccidial efficacy [37,55]. Nevertheless, several factors may explain the lack of significant differences in performance between experimental groups as observed in this study, such as the insufficient challenge due to oocyst concentration, vaccine type, housing conditions, and management practices. In line with our findings, Conway et al. (1999) reported that weight gain in infected birds was either unaffected or only slightly depressed (≤14 g) when exposed to a dose of 10⁴ oocysts per bird [37]. However, Fetterer and Allen (2000) observed a notable reduction in body weight gain in chickens challenged with 5 × 10⁵ oocysts per bird [55]. In our trial, we found that a dose of 4.0 × 10⁴ oocysts/bird did not significantly impact body weight gain. Furthermore, no treatment effect was observed, and feed conversion rates were similar across treatments. Feed intake at day 23 (993.8 ± 28.9 g) was lower than expected for Ross 308 (1383 g) [29], which may be attributed to the reduced lighting compared to standard commercial programs. Another crucial factor influencing the effectiveness of the vaccine challenge is the housing conditions. In this case, the birds were kept in battery cages. According to the technical datasheet of Coccivac^®^ D2 [56], pullets raised in battery cages can only be effectively immunized with Coccivac^®^ D2 if they are placed on litter no later than 4 days after vaccination. Humidity is essential for the sporulation of oocysts. If litter conditions are excessively dry during the immunization period, it is recommended to lightly spray it with water to facilitate the process. It is evident that the challenge protocol used in this case was different. However, this probably led to a less intense coccidial challenge. It is also important to highlight that the chosen vaccine type is not highly pathogenic. As indicated by Mathis et al. (2018), even at a 40× manufacturer-recommended dose, the non-attenuated Coccivac^®^ D2 vaccine resulted in an average lesion score of 1.7 for *E. tenella*, with only one out of ten birds exhibiting a score of three on the Johnson and Reid scale. Despite the lack of attenuation, the vaccine does not demonstrate high pathogenicity [57]. Additionally, the trial’s highly hygienic experimental conditions, including low stress, optimal density, and good management, may have facilitated the birds’ ability to overcome infection without compromising performance. Environmental factors such as stocking density and stress levels are crucial for detecting performance responses to plant-based feed additives [58].

## 5. Conclusions

The present study revealed that the tested phytogenic supplement was effective in reducing the fecal shedding of *Eimeria* spp. oocysts and intestinal lesions. It is a promising anticoccidial alternative compared to anticoccidial drugs to control coccidiosis in poultry without negatively affecting their productive performance. As the global population grows and the challenge of ensuring food security intensifies, enhancing the understanding and safe management of economically important livestock pathogens such as coccidia remain crucial. Future research should evaluate long-term effects and assess product efficacy and economic feasibility under commercial production conditions.

## Figures and Tables

**Figure 1 animals-15-00847-f001:**
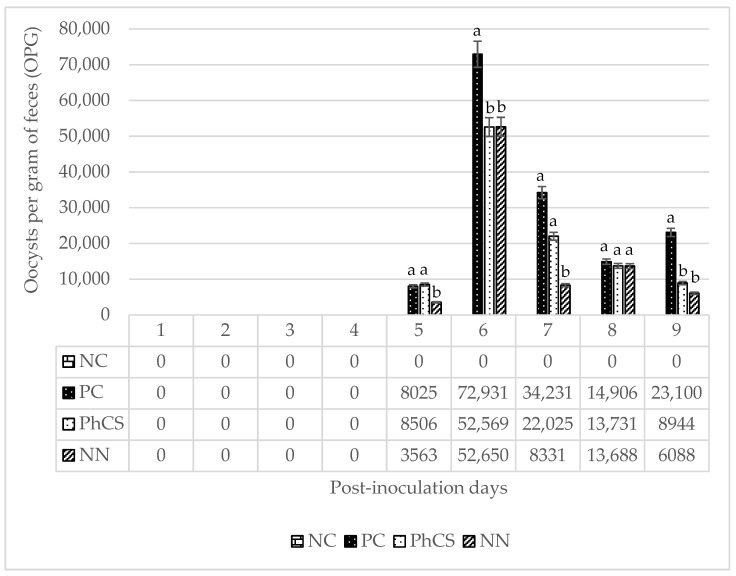
Fecal oocyst excretion per gram of feces (OPG) of *Eimeria* spp. from day 5 to 9 post-inoculation. Within each day, mean values with columns labeled with different letters differ significantly at *p* < 0.05. NC: negative control; PC: positive control; PhCS: phytogenic supplement; NN: anticoccidials (narasin and nicarbazin). Error bars represent a ±5% error. OPG: oocysts per gram of feces.

**Figure 2 animals-15-00847-f002:**
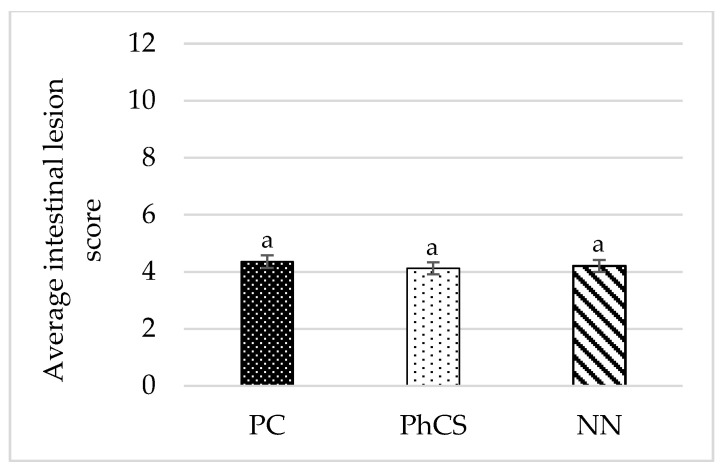
Average intestinal lesion score at 9 days post-inoculation. Mean values with columns labeled with the same letter do not differ significantly (*p* > 0.05). PC: positive control; PhCS: phytogenic supplement; NN: anticoccidials (narasin and nicarbazin). Error bars represent a ±5% error.

**Figure 3 animals-15-00847-f003:**
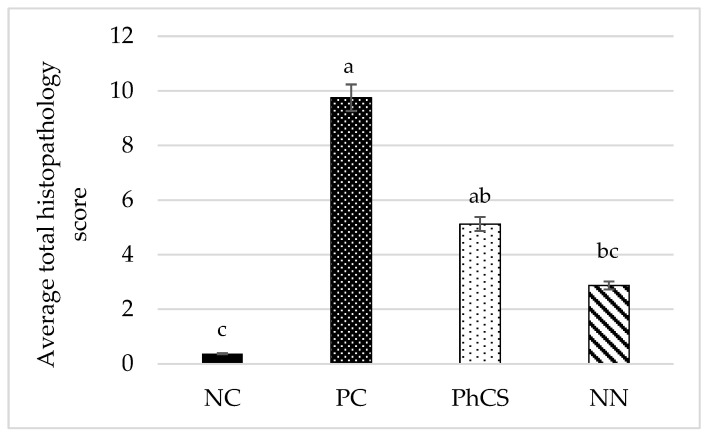
Average total histopathology score (sum of coccidia score and inflammation score) at 9 days post-infection. Mean values with different letters differ significantly (*p* < 0.05). NC: negative control; PC: positive control; PhCS: phytogenic supplement; NN: anticoccidials (narasin and nicarbazin). Error bars represent a ±5% error.

**Figure 4 animals-15-00847-f004:**
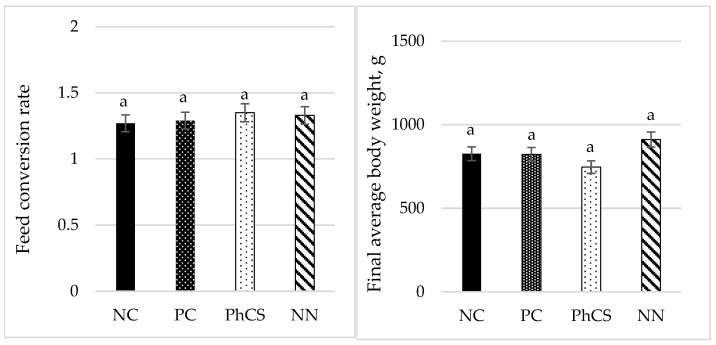
Final body weight and feed conversion rate at 9 days post-inoculation. Mean values with same letters differ significantly (*p* < 0.05). NC: negative control; PC: positive control; PhCS: phytogenic supplement; NN: anticoccidials (narasin and nicarbazin). Error bars represent a ±5% error.

**Table 1 animals-15-00847-t001:** Overview of experimental groups.

Experimental Group	Abbreviation	Coccidial Challenge	Supplementation
Negative control	NC	No	No
Positive control	PC	Yes	No
Phytogenic compound supplement	PhCS	Yes	Eimex^®^ at 1 kg/ton of feed, for 23 days
Anticoccidial	NN	Yes	Narasin and nicarbazin at 0.6 kg/ton of feed, for 23 days

**Table 2 animals-15-00847-t002:** Average total histopathology score of experimental groups, as the sum of the coccidia score and inflammation Score. NC: negative control; PC: positive control; PhCS: phytogenic supplement; NN: anticoccidials (narasin and nicarbazin).

Experimental Group	Coccidia Score	Inflammation Score	Average Total Histopathology Score
NC	0	0.375	0.375
PC	8	1.750	9.75
PhCS	3	2.125	5.125
NN	1.875	1	2.875

## Data Availability

The raw data supporting the conclusions of this article will be made available by the authors on request.

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
