# Peer review of "Evaluation of the Effectiveness of a Phytogenic Supplement (Alkaloids and Flavonoids) in the Control of Eimeria spp. in Experimentally Challenged Broiler Chickens"

_animals, 2025, doi:10.3390/ani15060847_

Round 1
Reviewer 1 Report
Comments and Suggestions for Authors
ln 53-57: English should be revised.
ln 60-71: the chapter on the mechanisms of resistance development is a bit off topic for the scope of the article. I would include this in a more specific study.
ln 94-96: the authors do not mention why the literature on the topic is unconvincing.
ln 212: the authors used a paired t-test to evaluate the changes of the OPG over time. the statistics presented in figure 1 is more likely to be an evaluation within timepoint and not over time. Also, t-test are used to evaluate 2 measurements, but in the manuscripts the groups were 4 and the days of OPG measurements were 5. Statistics should be revised throughly.
The captions below the figures are not clear. What letters represent should be stated more clearly (especially in figure 1, where it is not stated if the difference is within the timepoint)
Intestinal lesion scoring
The authors obtained a score by summing the scores of all the intestinal tracts. A separate analysis would provide interesting data on the different tracts, and the different Eimeria species that can potentially be affected. However, no significant difference was observed. Rather than the short trial period, I it possible that lesion score was assessed too late to observe differences. Most lesions appear during the active replication of the parasite in the intestine, so between day 4 and 7 pi, then the healing process begins. Lesion scores assessed as they are in the manuscript are not very representative of the anticoccidial power of the tested treatments.
Histopatological analysis and coccidia score
In the materials and methods section no description nor reference was cited for the preparation of the samples (fixation etc..). Also, the scoring system proposed by the authors is very interesting. However it would increase the reliablility of the results if they could include a reference of such method or explain the rational of the scores. Also, at line 184, they say that "coccidia" are counted, but no detail is given on the stage they are counting (oocysts, schizonts, ..?).
Errors
Errors are represented either as standard error or as 5% error. Please, be consistent with the errors.
Oocysts counts
the authors have included in the statistical analysis the negative control, which has no value because it was clean as expected. The real comparison is with the positive control, so the authors could exclude that group from the analysis.
Discussion section
The discussion section presents various citations to other research, but the author's results are not well discussed or included within the section. A brief paragraph on the results of the paper is reported at the end of the section, but it is not enough compared to the rest of the discussion.
Comments on the Quality of English Language
The english is acceptable but in some sections it should be improved. For example, in some passages it is not clear what article the authors are referring to (ie line 346: "the present study" is misleading if they are referring to another article).
In other parts (especially the introduction), the use of english denotes that the authors are not native speakers. I suggest an overall revision of the language to improve the fluency of the manuscript.
Reviewer 2 Report
Comments and Suggestions for Authors
Dear authors,
I've carefully revised your manuscript.
First of all, congratulations for the work you've done. Your manuscript deals with an important topic in Veterinary Parasitology, which is the search for sustainable solutions for parasite control.
Your research is of notewothy quality. However, several aspects need to be revised. All my improving suggestions can be easily viewed in the attached PDF file, using the "Comments" tool of Adobe Acrobat Reader. Your manuscript needs some minor revisions in the "Introduction" section, and major revisions in the "Materials and Methods", "Results" and "Discussion" sections.
Good luck for the rest of the publication process.
Best regards

Reviewer 3 Report
Comments and Suggestions for Authors
The manuscript addresses an important topic regarding alternative treatments to anticoccidials and is highly relevant. Minor revisions are needed, particularly in providing more details about the composition of the phytogenic additive being studied.
Title: It would be helpful to specify what the phytogenic supplement is in the title.
Simple Summary: The phrase “worked as well as the drugs” should be refined to: "showing comparable effects to anticoccidial drugs in controlling coccidiosis."
Abstract
Line 27-32: Mention the number of replicates for each group. What was the dosage of oocysts, and how was it administered (oral gavage)? Also, specify what the phytogenic supplement is.
Line 33-34: Mentioning the sample collection days for fecal samples would improve the abstract.
Line 34-36: Including the percentage reduction would make the abstract stronger.
Line 37-38: The phrase "intermediate protection based on histopathological scores" needs more context, such as the exact change in scores compared to the positive control, along with statistical relevance.
Introduction:
Line 58-69: The explanation of resistance mechanisms is good, but it needs more detail on how these mechanisms specifically relate to coccidiostats.
Line 87-105: The introduction on plant-derived anticoccidials is useful, but it should present a clearer research gap: "Despite the availability of botanicals with anticoccidial properties, their effectiveness varies across studies and needs further validation under controlled conditions." Including specific results from studies on the phytogenic additives would strengthen the introduction. Additionally, information on the composition of the phytogenic additive used in the current study should be included.
Line 104-105: Need to elaborate on these compounds and studies, including the composition and percentage of each constituent, as they are the key constituents in your phytogenic supplement, Eimex®. It would also be helpful to mention the source of extraction or how they were formulated.
Materials & Methods:
Line 113-116: What is the experimental design? Is it a randomized block design? What are the standard conditions? The temperature, humidity, and lighting conditions should be clearly stated for full transparency.
Line 118-119: The composition of the standard diet should be included in the main text or in a supplementary table.
Line 129-132: The composition of Eimex® (isoquinoline alkaloids, polyphenols) should be explained in more detail, including the concentration of the bioactive compounds.
Line 140-145: The reason for using a 20x standard dose should be explained, along with how it compares to infection levels seen in the field
Line 149-150: How many samples were collected? The methods mention collecting from all birds in each group. Do you mean samples were taken from each cage, or were samples pooled from each cage in a group before counting? Were replicates involved? Please provide more details.
Line 161: How were they euthanized—by cervical dislocation or another method, such as asphyxiation? Include how many chickens per group were used for the lesion assessments.
Line 172-173: Were the chickens randomized into each group for histopathological examination?
Line 205-212: The lesions and histopathology results should be analyzed using a non-parametric method. Since the scores for these data are not normally distributed, they should be analyzed with tests like the Kruskal–Wallis test.
Results
Line 216-226: Statistical differences are reported, but the percentage reduction or log values in oocyst shedding should also be provided.
Figure 1: The line numbers appear to be randomly scattered across the figure.
Line 247-272: Include the lesion and histopathology score values when explaining the results to strengthen the analysis.
Figure 4: Why is the feed intake graph not included?
Discussion
Line 333-334: The sentence should reflect antiviral activity rather than antibacterial, as the study focuses on Eimeria.
Line 346-349: Would the results be different if a cocktail of Eimeria strains was not used? Elaborate on the probable findings, compare them with existing literature, and provide justification.
The discussion should address why there was no improvement in growth performance, especially since previous studies have shown positive results. Compare these findings with anticoccidial treatments and explain the differences. Additionally, since the composition of the supplement is not provided, the comparison with polyphenols and alkaloids in the literature is challenging. This makes it difficult to pinpoint the exact mechanism, as the bioactive compounds may have different roles.
Any information on the economic evaluation and comparison with commercially used anticoccidials, along with the limitations of the study and suggestions for future research, would be helpful.
Reviewer 4 Report
Comments and Suggestions for Authors
As this study assessed various aspects of coccidial infection after challenge, the challenge model used is pivot to the study findings (results, discussion, conclusion). It is therefore of benefit to this paper to include more details about the challenge model. MSDs D2 vaccine contains non-precocious strains (unattenuated) which should be stated in the Methods section and raised in the context of the Discussion section. Recommendation to include any references supporting this challenge model and why it was expected to significantly affect BWt and FCR? If there isn't a supporting reference for the 20x dose, the authors could consider referencing Mathis etal 2018 "Comparison of breeder/layer coccidiosis vaccines: Part 1 - precocity and pathogenicity" which assessed the effect of inoculating a 40x dose of D2 vaccine in 7 day old and BWt + lesion scoring at 14do. That particular study concluded, 'Despite a lack of attenuation, the vaccine is not highly pathogenic as the 40x dose only produced an average lesion score 7dpc of 1.7 (for Eten).
It would also be helpful to include confirmation that the vaccine was applied within the expiry date (or how many months prior to the expiry date would be more helpful). As coccidosis vaccines are not preserved (liquid presentation held at 2-8C) viability declines with age. There is a long lead-time between individual antigen production, vaccine formulation and quality control + distribution, therefore live Eimeria vaccines are often available for use several months after manufacture. The rate of shelf-life decline is accelerated when combined with the effects of harsh chemical treatment processes during manufacture. Together these can reduce the clinical effects when using vaccines for challenge in coccidiosis studies. Therefore the challenge model used here (20x dose at 14do) should ideally be explained in terms of a reference or unpublished results from the group or even recommendation from an expert, together with the shelf-life of the vaccine, to align with the failure to induce BWt suppression within 9dpc (as reported by Mathis with a 40x dose).
Row 165: there were no reported mortalities during the study, and as the lesion scoring was performed for all birds at 23doa (9dpc), suggest removing 'death' from a score of 4.
Row 223: suggest replacing 'strong' with numerical. PhCS was nsd from the PC on days 5,7 and 8 (only d6 and d9), but the counts were numerically less than PC on each of these days. Therefore 'strong' is not an appropriate description for the difference between PhCS and PC 'between days 6-9'
Formatting: row numbers have jumped to overlap into Fig1 (editing required)
Row 250: there being nsd between the average intestinal lesion scores between all challenged groups, and Fig2 presents these results as comparable, suggestion to remove the sentence stating PhCS showed the 'lowest average score'.
Row271-272: the histo scores describe a combination of parasitism and host response (tissue damage and inflammation). The term 'protection' should be replaced with a more accurate description in relation to the histopathology scores. Additionally, there was nsd between the PhCS and PC, or PhCS and NN, therefore the average total histopathology score was numerically between the PC and NN scores but nsd from either.
General comment on Fig4: if other sections are going to make comment about numerical results that are not significant between treatment groups, it would be appropriate to include a statement in section 3.4 that the PhCS group had numerically the lowest final average BWt and highest final FCR of all the treatment groups (which may reflect the effects of the product under test, and not a reflection of the effects of coccidiosis challenge)
Discussion: last paragraph (from row 363) should include discussion of the challenge model and results found against other applications of the D2 vaccine challenge (especially the findings reported by Mathis 2018 as mentioned above).
The rest of the discussion and conclusions provide a very good overview of the use and previous findings of Phytogenics and their potential benefits to controlling coccidiosis (backbone of this study) which is the most important aspect of this research.
Comments on the Quality of English Language
English quality (spelling and grammar) is very good.
Round 2
Reviewer 2 Report
Comments and Suggestions for Authors
Dear authors,
Thank you for addressing my previous comments, and for all responses provided.
I've performed a second revision in your manuscript, which in my opinion has improved substantially. Congratulations.
However, some of my previous comments regarding the materials and methods, results, and references, were not addressed.
Please check the attached PDF file with my comments.
Thank you.
Best regards
